# Effects of the Health-Awareness-Strengthening Lifestyle Program in a Randomized Trial of Young Adults with an At-Risk Mental State

**DOI:** 10.3390/ijerph18041959

**Published:** 2021-02-18

**Authors:** Ching-Lun Tsai, Ya-Wen Lin, Hsing-Chi Hsu, Mei-Ling Lou, Hsien-Yuan Lane, Cheng-Hao Tu, Wei-Fen Ma

**Affiliations:** 1Department of Public Health (in Epidemiology and Preventive Medicine), China Medical University, Taichung 406040, Taiwan; chris.tsai70@gmail.com; 2Department of Public Health (in Nursing), China Medical University, Taichung 406040, Taiwan; wen5001@mail.cmu.edu.tw (Y.-W.L.); chiqueens@gmail.com (H.-C.H.); lahoya@hk.edu.tw (M.-L.L.); 3School of Nursing, Hungkuang University, Taichung 433304, Taiwan; 4Graduate Institute of Biomedical Sciences, China Medical University, Taichung 406040, Taiwan; hylane@gmail.com; 5Department of Psychiatry, China Medical University Hospital, Taichung 404332, Taiwan; 6Department of Psychology, College of Medical and Health Sciences, Asia University, Taichung 41354, Taiwan; 7The Brain Disease Research Center, China Medical University Hospital, Taichung 404332, Taiwan; 8Graduate Institute of Acupuncture Science, China Medical University, Taichung 404333, Taiwan; 9Program for Health Science and Industry, China Medical University, Taichung 406404, Taiwan; 10School of Nursing, China Medical University Hospital, Taichung 406040, Taiwan; 11Department of Nursing, China Medical University Hospital, Taichung 404332, Taiwan

**Keywords:** at-risk mental state, health awareness, health promotion lifestyle program, high risk for psychosis, prodrome

## Abstract

Background: Increasing health awareness in health promotion is considered as one of the less stigmatized interventions for improving help-seeking behaviors and total well-being. This study aimed to explore the short-term and long-term effectiveness of the health-awareness-strengthening lifestyle (HASL) program on Taiwanese young adults with at-risk mental state. Methods: A pre- and post-test randomized trial was conducted on 92 young adults with at-risk mental state. The HASL program was provided to the experimental group as intervention, and it was only provided to the control group passively by request after the post-test for ethical reasons. The program was conducted once every six weeks, 60–90 min per session, for a total of three times. Mental health risk, anxiety level, health promotion lifestyles, quality of life, physiological index, and physical exercises were assessed one week before and after the program for both groups and followed up to 6 and 12 months for experimental group only. Results: Compared to the control group, those in the experimental group showed significant improvements regarding anxiety level, health promotion lifestyles, and quality of life one week after participating in the program. Furthermore, the experimental group also showed an additional long-term positive effect on mental risk, physical exercises, and physical health after the follow-ups. Conclusions: The outcomes highlighted the interventions of the HASL program leading to more positive health effects on young adults with at-risk mental state. The implementation of similar clinical service is recommended for young adults with at-risk mental state.

## 1. Introduction

Mental illness can cause a tremendous long-term burden on individuals, families, and society. In the 66th World Health Assembly, the World Health Organization (WHO) adopted a comprehensive mental health action plan that focused on the international mental health issues and human rights [1]. The health promotion in mental health, especially in relation to early detection and intervention, has become a key global health issue [2]. The screening of individuals at high risk for psychosis has attracted increasing attention from international clinical researchers [3]. In the past decade, there have been many studies on individuals at high risk for psychosis focused on different areas, for instant, the concepts of at-risk mental state (ARMS) [4] and ultra-high risk [5], and the possible treatments [6]. Those individuals mostly raised their concerns on attenuated positive symptoms, family issues, and social problems [6].

The concept of ARMS or ultra-high risk enables the clinical diagnosis of high risk for psychosis [5]. It is characterized internationally by the presence of one or more psychotic symptoms, including ideas of reference, odd beliefs, paranoid ideation, perceptual disturbance, and schizotypal personality [4,5]. In this study, the concept of ARMS emphasized more on negative symptoms and anxiety. This was because negative symptoms, such as schizotypal traits, are statistically predictive factors for the transition from ARMS to mental illness [7,8], and high levels of perceived stress and anxiety also occur more frequently in individuals with ARMS than in the general public [8,9]. Those individuals had higher risk of developing a first episode of psychosis within one to two years [4]. Moreover, 18–36% may progress to full psychotic disorder within three years [10]. Thus, individuals with ARMS are the population particularly needing professional help both physically and psychologically [11,12].

The intervention goals for people at high risk for psychosis are generally to reduce current symptoms and the risk of developing a full psychosis [12]. Some interventions also include secondary outcomes, such as improving individual functioning and reducing the risk of possible comorbidities [4,13]. Traditional interventions are usually carried out through psychotherapy (e.g., cognitive behavioral therapy, family therapy), pharmacotherapy (e.g., olanzapine, risperidone, dietary supplements), or a combination of both [6]. Psychopharmacological medications can target specific symptoms; however, they are often associated with noticeable side effects and social stigma [14]. Thus, novel and safe treatment interventions or strategies are in high demand.

Health promotion is considered to be a vital strategy for improving health and managing symptoms [15]. According to the WHO (2019), health promotion is an activity that encourages people to adopt and maintain healthy lifestyles, and it creates supportive living conditions and healthy environments. It includes positive activities that increase individual well-being and actualize health potential [1]. Many studies have included self-initiated health promotion lifestyle strategies, such as self-responsibility, stress management, interpersonal support, exercise, and nutrition to improve the quality of life for both patients with chronic disorders and individuals with psychiatric disorders [16,17]. The relationships of physical disease, depression, and health-promoting lifestyles also had strongly regarded the importance of health-promoting lifestyles in individuals with psychiatric disorder due to facilitating changes in total health [17]. Furthermore, Parker et al. (2011) found that physical activity can reduce depressive symptoms in 15–25-year-old adolescents [18].

People with ARMS tend to have high rates of unhealthy lifestyle factors, such as physical inactivity [11]. Carney et al. (2016) emphasized physical health promotion strategies for people with ARMS, stating that possible methods, such as education, persuasion, and training, provide a systematic approach for developing interventions to promote behavioral change [11]. In addition, strengthening one’s health awareness is in a form of education to enhance help-seeking behaviors and improve well-being. Furthermore, it provides knowledge relevant to mental health symptoms and treatment [19]. Increasing health awareness is even more crucial for those at high risk for psychosis, as their condition as not yet developed into a psychotic disorder. This study aimed to explore the short-term and prolonged effects of the health-awareness-strengthening lifestyle (HASL) program, which is a health promotion lifestyle program that emphasizes self-awareness of mental health risk for the young Taiwanese adults with ARMS.

## 2. Methods

### 2.1. Research Design

A randomized pre- and post-test control group experimental design was employed for this study. The research flowchart is presented in Figure 1.

### 2.2. Study Participants

The purposive sampling was used to recruit participants between the ages of 20 and 35 years from university counseling centers, outpatient psychiatric clinics, and community counseling centers. Three scales were used to identify individuals with higher risk for developing psychosis, including the Chinese Version of the Schizotypal Personality Questionnaire-Brief (CSPQ-B), Chinese Mandarin State and Trait Anxiety Inventory Form Y (CMSTAI-Y), and a demographic inventory to collect information on participants’ family history of mental illness.

Inclusion criteria:

Ages of 20–35 years old males or females.

Having one of the following conditions:

Scored ≥ 17 on the CSPQ-B [8,20].

Scored ≥ 60 for trait anxiety on the CMSTAI-Y [8,21].

Reported a family history of mental illness in a lineal blood relative, plus either a score of 15–16 on the CSPQ-B or a scored of greater than 60 for state anxiety on the CMSTAI-Y [8,20,21].

Capacity and willingness to give written informed consent.

Exclusion criteria:

Confirmed diagnosis of DSM-5 criteria for schizophrenia, mood disorders, substance use disorder, or other psychotic disorders [22].

### 2.3. Sample Characteristics

A total of 92 young adults with ARMS participated in the study. Their ages ranged from 19 to 26 years with a mean age of 21.34 years (SD = 1.28). The majority of the participants were female 51 (55.4%), and 55 (59.8) reported that they were religious. Very few reported smoking or alcohol consumption habit: 2 (2.2%) and 5 (5.4%), respectively. There were 34 (37%) participants who reported experiencing sleep disturbance in the past three months; however, only 1 (1.1%) reported taking hypnotic medications. Among the 92 study participants, 16 (17.4%) reported previously seeking help for mental health issues and 29 (31.5%) reported having a family history of mental illness.

### 2.4. Data Collection

The data of this study were collected from October 2014 to July 2018. The participants were randomly assigned by computer coding to either the experimental or control group directly after they were recruited, provided informed consent, and completed the pre-test. The experimental group was provided with the HASL program. The assessments were done one week before (T0) and after (T1) the program to evaluate its short-term effects. The program would only be provided to the control group passively, by request, after the post-test. The experimental group was followed up with at 6 months (T2) and 12 months (T3) after the program to assess its prolonged effects.

### 2.5. Health-Awareness-Strengthening Lifestyle Program

The HASL program is a lifestyle program comprising three main elements: exercise, nutrition, and health responsibility. It is derived from the six dimensions of health-promoting lifestyle: exercise, nutrition, stress management, interpersonal support, self-actualization, and health responsibility [23]. The one-on-one program was conducted once every six weeks, for 60–90 min per session, a total of three times.

The first session focused on the increasing exercise as the main mechanism for stress adjustment. The researcher would first establish a trusting relationship with participants and later discuss with the participants their current main stressors, introduce regular aerobic exercise for enhancing stress resistance, and develop an exercise plan.

The second session included discussion of participants’ difficulties with maintaining the exercise plan. The researcher would also bring in the concept of balanced nutrition to further enhance participants’ healthy lifestyles. The last session included knowledge and concepts related to mental health risk, strengthened participants’ understanding of responsibility for their own health, encouraged self-monitoring, and cultivated responsible attitudes toward health. Health education pamphlets/manual and a personal reflecting journal were used during the sessions. The detailed procedures are shown in Table 1.

### 2.6. Instruments and Measurements

#### 2.6.1. Demographic Inventory

The demographic inventory includes the self-reported personal information on gender, age, religion, smoking habits, alcohol consumption habits, sleep disturbance over the past three months, previous mental health history, and family mental health history.

#### 2.6.2. Chinese Version of the Schizotypal Personality Questionnaire-Brief

The Schizotypal Personality Questionnaire-Brief (SPQ-B) is a 22-item questionnaire developed by Raine and Benishay (1995). It comprises three aspects of deficits: eight questions for cognitive–perceptual, eight for interpersonal, and six for disorganized. Higher scores indicate greater degrees of deficit [24]. The SPQ-B was translated by Ma et al. (2010) into the CSPQ-B and tested on 618 undergraduate students. They found an internal consistency of 0.76 and two-week test–retest reliability of 0.82. The sensitivity and specificity were 80.0% and 85.9%, respectively, in identifying undergraduate students’ susceptibility to psychosis. Furthermore, the optimal cut-off score was found to be 17 [20].

#### 2.6.3. Chinese Mandarin State and Trait Anxiety Inventory Form Y

The State and Trait Anxiety Inventory Form Y (STAI-Y) was modified from the original Form X by Spielberger et al. (1983) and translated by Ma et al. (2013) into the CMSTAI-Y. It consists of 20 items each for state anxiety and trait anxiety rated on a 4-point Likert scale. Higher scores indicate higher degrees of anxiety; a score between 60 and 80 points indicates a high level of anxiety [25]. The CMSTAI-Y has been tested on 306 Taiwanese adults with anxiety disorders. Cronbach’s α for the internal consistency of the state and trait anxiety subscales were 0.91 and 0.92, respectively. The two-week test–retest reliabilities were 0.76-0.91. The high correlations between the CMSTAI-Y and the Chinese Hamilton Anxiety Rating Scale (r = 0.69 for state and 0.74 for trait anxiety) indicated good criterion validity [21].

#### 2.6.4. Health-Promoting Lifestyle Profile–Short Form

The Health-Promotion Lifestyle Profile—Short Form (HPLP-S) was revised to be suitable for Chinese-speaking respondents by Wei and Lu (2005), based on the original HPLP [23]. This 24-item questionnaire includes six subscales on stress management, self-actualization, health responsibility, interpersonal support, exercise, and nutrition, covering the physical, psychological, and social levels of the concept of “health behavior” [26]. Higher scores indicate better performance on health-promoting lifestyle behaviors. Wei and Lu (2005) tested the HPLP-S with a sample of 967 college students and found that the internal consistency coefficient was 0.90 for the total scale and ranged from 0.63 to 0.79 for the subscales. The correlation coefficients between the subscales of the HPLP and HPLP-S all reached the level of significance (*p* < 0.001). Good construct validity and convergent validity was supported [26].

#### 2.6.5. World Health Organization Quality of Life-Brief Taiwan Version

The World Health Organization Quality of Life-Brief (WHOQOL-BREF) was derived from the original WHOQOL-100 [27]. It is a simplified, cross-cultural version designed for generic use. The Taiwan version (WHOQOL-BREF TW) consists of 28 items (26 items from the original WHOQOL-BREF plus 2 region-specific/national items) and measures four domains: physical, psychological, social relationships, and environmental [28]. Higher domain scores indicate a better quality of life. The 2-to-4 week test–retest reliability coefficients ranged from 0.76 to 0.80 at the domain level (*p* < 0.01) and 0.41 to 0.79 at the item level (*p* < 0.01). The internal consistency coefficients for the four domains ranged from 0.70 to 0.77. The content validity coefficients ranged from 0.51 to 0.64 for inter-domain correlations (*p* < 0.01) and 0.53 to 0.78 for item–domain correlations (*p* < 0.01) [28].

#### 2.6.6. 3-Month Physical Activity Checklist

The 3-Month Physical Activity Checklist (3MPAC) was developed by Ma et al. (2011). It is a self-report scale with 18 items measuring the type, frequency, and intensity of physical activity in the past three months for adults with psychiatric disorders. Its test–retest reliability ranged from 0.71 to 0.86 for light, moderate, and heavy exercises [29]. The efficacy was tested on 98 adults with schizophrenia, 22 adults with bipolar disorder, and 153 adults with anxiety disorders by criterion validity testing with a 7-Day Physical Activity Recall interview (r = 0.47 for light, r = 0.64 for moderate, and r = 0.73 for heavy exercise) and cross-sample testing (χ^2^ = 21.98, *p* < 0.000; Ma et al., 2011). Ma et al. (2017) also used the 3MPAC to assess 83 patients with anxiety disorders with a mean age of 40.11 in Taiwan [29].

#### 2.6.7. Physical Assessments

The participants’ systolic blood pressure (BP), diastolic BP, height and weight for calculation of body mass index (BMI), waist circumference, and hip circumference were measured by non-invasive approaches.

### 2.7. Data Analysis

SPSS for Windows version 22.0 (IBM, Armonk, NY, USA) was used for data archiving and statistical analysis. The descriptive and inferential statistical analysis consisted of two sample independent t-test, chi-square test, Fisher’s exact test, and generalized estimating equation (GEE). The statistical significant level was set at *p* < 0.05.

### 2.8. Identifying Information and Ethical Considerations

This study confirms that all methods were performed in accordance with the relevant guidelines and regulations. This study was reviewed and approved by the ethical review board of China Medical University Hospital, Taiwan (No: DMR101-IRB2-222, DMR101-IRB2-22(CR01), DMR101-IRB2-222(CR02), CMUH104-REC3-114, and CMUH104-REC3-114(CR-1)) to assure all the ethical requirements and standards were strictly followed. All eligible participants who met the inclusion criteria were asked to provide a signed informed consent. The collected questionnaires were stored and locked in the office to ensure data confidentiality. Only participants’ ID codes were used throughout the study. Data were not used by a third party. Participants could withdraw at any time without penalty, and their decision would not affect their rights to seek medical treatment or schooling. Participation in the study was entirely voluntary, and anonymity was highly guaranteed.

## 3. Results

### 3.1. Study Variables

The participants were randomly assigned into either the experimental group (*n* = 46, 50.0%) or control group (*n* = 46, 50.0%). There were significant differences in age (t = −2.8, *p* = 0.0061) between the two groups, but the mean ages were closed between groups: 21.7 (SD = 1.5) and 21.0 (SD = 0.9) in the experimental and control groups, respectively. 

Table 2 shows the differences in study variables between the two groups for the pre-test. Only the nutrition dimension of the health promotion lifestyles appeared to be significantly higher in the experimental group during the pre-test (t = −3.1, *p* = 0.0028). We found no other significant differences in study variables between the two groups for the pre-test.

### 3.2. Effects of the HASL Program in Post-test

After participating in the HASL program, the experimental group had significantly higher scores for overall health promotion lifestyle (t = −2.7, *p*= 0.0075), the nutrition dimension of the health promotion lifestyle (t = −3.2, *p*= 0.0017), overall quality of life (t = −2.3, *p*= 0.0230), and the psychological aspect of quality of life (t = −2.3, *p* = 0.0237). This group also showed significantly lower scores for state (t = 2.8, *p* = 0.0060) and trait anxiety (t = 2.7, *p* = 0.0093). Additionally, the experimental group had significantly fewer participants with a score over 60 for state and trait anxiety, compared to the control group, and the p-values were 0.0217 and 0.0197, respectively. The results of the differences in study variables between the two groups for the post-test are also shown in Table 2.

### 3.3. Effects of the HASL Program in Differences between Pre- and Post-Test Analysis (T1–T0)

The differences between pre- and post-test analysis in the two groups were examined, due to a few differences in variables being found during the pre-test. In this analysis, the experimental group demonstrated significant improvements in the self-actualization dimensions of health promotion lifestyle (*t* = −2.2, *p*= 0.0291), as well as for state (*t* = 2.9, *p* = 0.0048) and trait anxiety (*t* = 3.2, *p* = 0.0021). The details are shown in Table 3.

### 3.4. Prolonged Effects of the HASL Program

Of the 46 participants in the experimental group, 38 (82.6%) completed the T3 analysis; 8 (17.4%) were too busy to continue the study. Most of the study variables showed significant differences through T0–T3. Compared with T0, overall health promotion lifestyle (*p* = 0.0007) showed significant improvement at T1. State anxiety only showed a significant reduction at T1 (*p* < 0.0001). Trait anxiety reduced significantly at T1, T2, and T3 when compared with T0 (*p* < 0.0001). Overall schizotypal personality also showed a significant decrease at T1, T2, and T3. Overall quality of life showed significant improvements at all three time points. Regarding physical assessments, only diastolic BP (*p* = 0.0176) at T1 and waist/hip ratio (*p* = 0.0190) at T2 showed significant differences. Furthermore, the overall level of physical exercise increased significantly over time. The differences in study variables between T0 and T3 in the experimental group are shown in Table 4.

## 4. Discussion

This study provided the HASL program to young adults with ARMS to enhance their mental health awareness and healthy lifestyle. The results showed that the HASL program had short-term effects for reducing overall anxiety, enhancing healthy lifestyle, and improving quality of life. Moreover, the HASL program also had long-term positive effects on reducing mental risk and anxiety, increasing physical exercise and healthy promotion lifestyle, and improving overall physical health and quality of life. The young adults with ARMS in the present study had not yet met the diagnostic threshold for a psychotic disorder; therefore, it was important to strengthen their awareness on mental health risks and increase their self-awareness, thus reducing their mental health risks and improving their holistic health. As a result, the HASL program was shown to be suitable for the current study population.

The study participants who attended the HASL program showed a significant short-term reduction in anxiety and long-term improvement in mental health risk. Taking anti-psychotic medications, such as risperidone, has the most significant effect on reducing or preventing psychotic symptom development during early episodes of psychosis [5,30]; however, the side effects, anxiety, and stigma associated with taking anti-psychotic medications can negatively impact individuals’ daily lives [31]. Conversely, psychosocial interventions usually cause less stigmatization [30,32]. Thus, in this study, the HASL program was shown to be acceptable for and provide substantial help to young adults with ARMS.

In addition to a healthy lifestyle including regular exercise, balanced nutrition, and stress management, the HASL program increased self-awareness and personal health awareness to strengthen the concept of self-health responsibility in the experimental group. In a previous study on high risk for psychosis, Schwingel and Gálvez [33] stated that the enhancement of personal health awareness should be included in interventions to reduce risk. The participants’ own sense of self-achievement and health responsibility were enhanced by these strategies, thus showing the program’s effects at long-term follow-up. The present study not only met the suggestion of Schwingel and Gálvez [33] but also provided empirical evidence to support the notion that health awareness can reduce mental health risk in young adults with ARMS. Furthermore, Olvet et al. [34] emphasized that although individuals with ARMS have not yet developed psychotic disorders, their symptoms have already affected their daily functioning. The findings of the study support this statement and suggest that knowledge and personal health awareness related to mental health should be included early in treatment interventions.

The HASL program not only showed the effects of reducing mental health risk in young adults with ARMS but also had positive effects on emotion, anxiety, and quality of life. Increasing physical exercise is an important element of the HASL program. The benefits of regular exercise as an effective stress management strategy [35] may help explain the reason for the improvements noted in the present study. Vélez-Toral et al. [36] also reported that their participants also experienced positive changes in their short- and long-term health after the Exercise Plus Health Promotion Intervention. In considering the provision of safe interventions for research participants, the HASL program showed positive long-term effects on overall physical, emotional, and mental health risk, as well as quality of life in young adults with ARMS. Other studies have found that implementing an exercise program as an adjunctive treatment can improve positive and negative symptoms, depression, anxiety, cognition, and quality of life in patients with schizophrenia [37]. Our results demonstrated that the participants did not experience positive effects other than short-term reduction in overall anxiety after increasing their amount of exercise. However, in the long-term follow-up, regardless of overall mental health risk, interpersonal interaction, cognition, thought organization, mood, and quality of life all showed statistically significant improvements.

The level of physical activity is often difficult to maintain unless there is strong motivation [38,39,40]. This study found that the participants’ amount of physical exercise in the HASL program was maintained even at the third follow-up. This might be as a result of the HASL program strengthening individual’s health responsibility for their self-awareness on mental health risk, thus further providing them with additional motivation to continue exercising regularly. This allowed the amount of exercise to increase significantly with the positive progress of mental health risk, mood, and quality of life at follow-up compared to the pre-test.

This study also had several limitations. First, the control group was not included in the long-term follow-up due to the ethical consideration that they might receive the HASL program passively, which could cause them to have similar results as the experimental group if we conducted the follow-ups. Therefore, it is uncertain to what extent their long-term outcomes might have showed change compared with those of the experimental group. Another limitation is the small sample size. Furthermore, future research could consider including more precise and objective physiological measures, such as blood serum examination or biomedical imaging. Lastly, we only evaluated the effects of the HASL program for young Taiwanese adults with ARMS. Due to age and cultural differences, the applicability of the HASL program in other populations needs to be further assessed in future research.

## 5. Conclusions

The HASL program was developed based on the elements of exercise, nutrition, and health responsibility. It emphasizes self-awareness of mental health risk, aims to reduce stress and anxiety, and tries to delay or avoid the onset of psychosis. Our results demonstrated that the HASL program had significant effects on reducing overall anxiety, mental health risk, and anxiety; promoting healthy lifestyles; increasing physical exercises; and improving quality of life. These findings can be the basis for clinical implementation of health promotion lifestyle programs on individuals with ARMS to improve their physically and psychological well-being before the development of psychotic disorder. The program should aim at delaying, reducing, or even avoiding the onset of disease to lower the subsequent long-term damages and social burdens.

## Figures and Tables

**Figure 1 ijerph-18-01959-f001:**
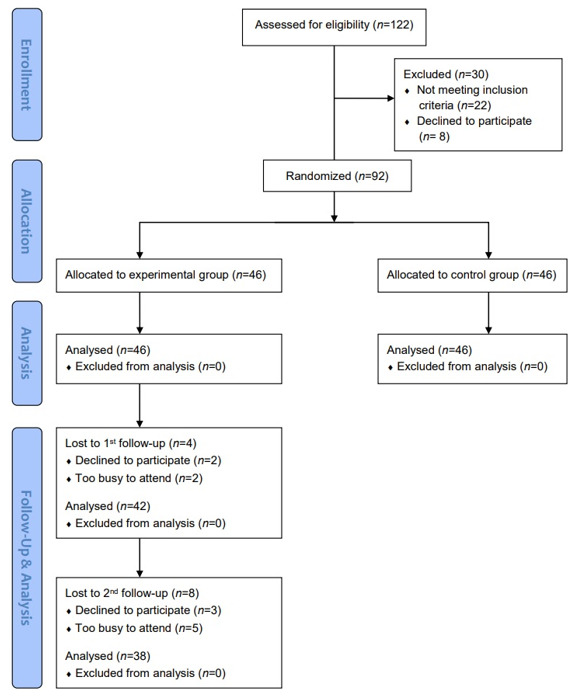
Research flowchart.

**Table 1 ijerph-18-01959-t001:** The detailed procedures of health-awareness-strengthening lifestyle program.

Session	Dimensions	Definitions *	Prevention Strategies
1	Exercise	Engaging in sports and leisure activities	Counseling: Establishing role model Encouraging sharing of previous sports experience Providing mechanisms linking exercise and stress management Tools: Health education manual on benefits of sports and the obstacles to overcome, as well as exercise journal
	Stress Management	Using relaxation techniques for stress management	Counseling: Advising time management and planning Encouraging the use of regular moderate aerobic exercise as the stress management ability Training of crisis management and response capabilities Tools: Stress management manuals
	Interpersonal Support	Developing social support systems and intimate relationships, encouraging sharing of ideas and spending time with friends and family, etc.	Counseling: Discussing the establishment of social network Training communication skills Strengthening mutual communication Encouraging the sharing of interpersonal difficulties and solution strategies
2	Nutrition	Balancing daily diet and food choice	Counseling: Encouraging sharing of balanced nutritional intake Tools: Nutrition manuals, nutrition knowledge triangle, and nutrition journal
3	Self-Actualization	Maximizing individual abilities and potentials, including purpose in life, self-appreciation, positive thinking, etc.	Counseling: Encouraging sharing of personal vision, planning, and performance in academic and living, etc. Tools: Journal
	Health Responsibility	Paying attention to health condition, seeking professional assistance if needed, and attending health lesson	Counseling: Encouraging sharing of health responsibility issues Establishing a health responsibility role model Tools: Mental health related manuals, emotional self-assessment booklet, and journal

* Walker, Sechrist, and Pender, 1987.

**Table 2 ijerph-18-01959-t002:** Differences in study variables between two groups in pre-test and post-test (two weeks after) analysis (N = 92).

	Pre-Test				Post-Test			
	Control (*n* = 46)	Experimental (*n* = 46)	*t*	*p*	Control (*n* = 46)	Experimental (*n* = 46)	*t*	*p*
	Mean (SD)	Mean (SD)			Mean (SD)	Mean (SD)		
Health Promotion Lifestyles	52.9 (9.4)	54.9 (7.8)	−1.1	0.2586	53.8 (8.4)	58.5 (7.8)	−2.7	0.0075
Self-Actualization	9.7 (2.5)	9.6 (2.1)	0.0	0.9638	9.6 (2.6)	10.5 (2.1)	−1.7	0.0897
Health Responsibility	7.0 (2.7)	7.1 (1.9)	−0.3	0.7538	7.3 (2.3)	7.9 (2.1)	−1.3	0.2010
Exercise	7.9 (2.0)	7.8 (2.1)	0.3	0.7996	7.7 (2.2)	8.4 (2.1)	−1.5	0.1397
Nutrition	9.0 (2.4)	10.4 (1.9)	−3.1	0.0028	9.4 (2.3)	10.9 (2.0)	−3.2	0.0017
Interpersonal Support	10.2 (2.8)	10.2 (2.5)	0.0	1.0000	10.3 (2.5)	10.6 (2.1)	−0.7	0.5134
Stress Management	9.1 (2.0)	9.7 (2.1)	−1.5	0.1347	9.5 (1.9)	10.3 (2.1)	−1.8	0.0694
State Anxiety	47.9 (6.7)	47.7 (8.8)	0.1	0.8834	47.7 (9.5)	42.5 (7.7)	2.8	0.0060
Trait Anxiety	56.6 (5.6)	55.7 (7.4)	0.7	0.5079	56.0 (7.0)	52.0 (7.0)	2.7	0.0093
Schizotypal Personality	11.0 (4.5)	11.4 (3.8)	−0.5	0.6158	10.3 (4.6)	9.9 (4.0)	0.4	0.6686
Cognitive−Perceptual Deficits	3.7 (1.9)	3.5 (1.4)	0.4	0.6612	3.5 (1.6)	3.0 (1.6)	1.3	0.1821
Interpersonal Deficits	5.0 (2.3)	5.5 (2.1)	−1.0	0.3440	4.6 (2.4)	4.8 (2.2)	−0.4	0.6795
Disorganization	2.3 (1.8)	2.4 (1.7)	−0.4	0.6782	2.1 (1.8)	2.0 (1.8)	0.4	0.7081
Quality of Life	53.9 (7.5)	56.0 (7.1)	−1.2	0.2165	55.0 (8.8)	59.3 (7.3)	−2.3	0.0230
Physical	12.6 (2.2)	12.8 (2.0)	−0.3	0.7582	13.2 (2.4)	13.4 (2.3)	−0.5	0.6067
Psychological	10.7 (2.4)	11.3 (2.0)	−1.3	0.2045	10.9 (2.5)	12.1 (2.3)	−2.3	0.0237
Social	11.7 (2.3)	12.4 (1.9)	−1.3	0.2019	12.3 (2.7)	12.9 (1.8)	−1.3	0.2060
Environmental	13.4 (1.9)	13.4 (1.9)	0.0	1.0000	13.6 (2.1)	14.2 (1.9)	−1.4	0.1726
Physical Assessments								
Systolic Blood Pressure	115.7 (15.7)	111.7 (16.1)	1.2	0.2383	118.6 (19.6)	111.8 (15.9)	1.8	0.0805
Diastolic Blood Pressure	74.0 (10.6)	70.2 (9.8)	1.8	0.0765	75.3 (10.8)	72.9 (10.9)	1.0	0.3145
Body Mass Index	22.5 (4.7)	22.5 (4.0)	0.0	0.9658	22.5 (4.8)	22.6 (4.0)	−0.1	0.9419
Waist/Hip Ratio	0.79 (0.06)	0.77 (0.08)	1.0	0.5384	0.80 (0.09)	0.78 (0.07)	1.3	0.2293
Physical Exercises	68.4 (94.9)	50.5 (95.6)	0.9	0.3695	105.1 (121.1)	94.0 (119.8)	0.4	0.6682

**Table 3 ijerph-18-01959-t003:** Differences in study variables between pre- and post-test analysis in two groups.

	Control (*n* = 46)	Experimental (*n* = 46)	*t*	*p*
	Mean (SD)	Mean (SD)		
Health Promotion Lifestyles	0.83 (6.82)	3.61 (7.32)	−1.8	0.0699
Self-Actualization	−0.02 (1.44)	0.83 (2.11)	−2.2	0.0291
Health Responsibility	0.26 (1.52)	0.76 (1.78)	−1.4	0.1622
Exercise	−0.17 (1.92)	0.57 (2.00)	−1.7	0.0841
Nutrition	0.21 (1.88)	0.48 (1.72)	−0.7	0.4938
Interpersonal Support	0.17 (1.82)	0.39 (1.77)	−0.6	0.5589
Stress Management	0.38 (1.86)	0.59 (1.97)	−0.5	0.6165
State Anxiety	−0.38 (8.70)	−5.13 (6.30)	2.9	0.0048
Trait Anxiety	−0.40 (4.28)	−3.72 (5.39)	3.2	0.0021
Schizotypal Personality	−0.69 (3.13)	−1.52 (2.87)	1.3	0.1966
Cognitive–Perceptual Deficits	−0.24 (1.65)	−0.48 (1.38)	0.7	0.4593
Interpersonal Deficits	−0.38 (1.50)	−0.61 (1.69)	0.7	0.5073
Disorganization	−0.07 (1.11)	−0.43 (1.56)	1.3	0.2089
Quality of Life	0.56 (4.69)	2.98 (6.06)	−1.9	0.0616
Physical	0.52 (1.36)	0.65 (2.03)	−0.4	0.7253
Psychological	0.14 (1.49)	0.77 (1.92)	−1.7	0.0934
Social	0.35 (2.32)	0.44 (1.52)	−0.2	0.8397
Environmental	0.19 (1.58)	0.82 (2.06)	−1.6	0.1153
Physical Assessments				
Systolic Blood Pressure	3.31 (12.52)	0.22 (9.67)	1.3	0.1997
Diastolic Blood Pressure	1.38 (8.81)	3.02 (8.33)	−0.9	0.3742
Body Mass Index	0.10 (0.71)	−0.01 (1.15)	0.5	0.6034
Waist/Hip Ratio	0.01 (0.06)	0.01 (0.05)	0.5	0.5968
Physical Exercises	46.03 (92.26)	43.52 (124.11)	0.1	0.9152

**Table 4 ijerph-18-01959-t004:** Tendency between T0 and T3 in the experimental group by generalized estimating equation analysis.

	T0 (*n* = 46)	T1 (*n* = 46)	T2 (*n* = 42)	T3 (*n* = 38)	*p*		
	Mean (SD)	Mean (SD)	Mean (SD)	Mean (SD)	T1 vs. T0	T2 vs. T0	T3 vs. T0
Health Promotion Lifestyles	54.9 (7.8)	58.5 (7.8)	57.3 (8.1)	56.6 (9.8)	0.0007	0.0584	0.1610
Self-Actualization	9.6 (2.1)	10.5 (2.1)	10.1 (2.1)	9.8 (2.2)	0.0073	0.0491	0.4144
Health Responsibility	7.1 (1.9)	7.9 (2.1)	7.5 (1.8)	7.4 (2.0)	0.0034	0.2591	0.4290
Exercise	7.8 (2.1)	8.4 (2.1)	8.0 (2.2)	8.2 (2.1)	0.0522	0.6554	0.4158
Nutrition	10.4 (1.9)	10.9 (2.0)	11.0 (2.2)	10.9 (2.1)	0.0569	0.1148	0.2357
Interpersonal Support	10.2 (2.5)	10.6 (2.1)	10.5 (2.5)	10.3 (2.3)	0.1295	0.3160	0.6366
Stress Management	9.7 (2.1)	10.3 (2.1)	10.2 (1.8)	10.1 (2.3)	0.0413	0.0782	0.1006
State Anxiety	47.7 (8.8)	42.5 (7.7)	46.0 (8.4)	45.6 (9.2)	<0.0001	0.2657	0.2201
Trait Anxiety	55.7 (7.4)	52.0 (7.0)	51.8 (7.6)	52.2 (8.3)	<0.0001	<0.0001	<0.0001
Schizotypal Personality	11.4 (3.8)	9.9 (4.0)	10.1 (4.3)	10.2 (4.5)	0.0003	0.0092	0.0030
Cognitive–Perceptual Deficits	3.5 (1.4)	3.0 (1.6)	3.1 (1.9)	3.2 (1.8)	0.0173	0.1374	0.2905
Interpersonal Deficits	5.5 (2.1)	4.8 (2.2)	4.8 (2.1)	5.1 (2.4)	0.0137	0.0151	0.0050
Disorganization	2.4 (1.7)	2.0 (1.8)	2.2 (1.7)	1.9 (1.5)	0.0558	0.0870	0.0023
Quality of Life	56.0 (7.1)	59.3 (7.3)	58.2 (7.4)	58.4 (9.1)	0.0019	0.0308	0.0115
Physical	12.8 (2.0)	13.4 (2.3)	13.5 (2.6)	13.7 (2.3)	0.0293	0.0258	0.0032
Psychological	11.3 (2.0)	12.1 (2.3)	11.7 (2.4)	11.8 (2.7)	0.0060	0.3133	0.1493
Social	12.4 (1.9)	12.9 (1.8)	12.9 (1.8)	12.7 (2.5)	0.0471	0.0489	0.2025
Environmental	13.4 (1.9)	14.2 (1.9)	13.8 (2.1)	14.1 (1.7)	0.0083	0.1534	0.0131
Physical Assessments							
Systolic BP	111.7 (16.1)	111.8 (15.9)	114.1 (16.5)	113.5 (17.7)	0.8858	0.4731	0.2516
Diastolic BP	70.2 (9.8)	72.9 (10.9)	72.7 (11.2)	73.2 (11.5)	0.0176	0.1590	0.0536
Body Mass Index	22.5 (4.0)	22.6 (4.0)	22.5 (4.1)	22.8 (4.8)	0.9655	0.6323	0.0844
Waist/Hip Ratio	0.77 (0.08)	0.78 (0.07)	0.80 (0.07)	0.78 (0.08)	0.4068	0.0190	0.0784
Physical Exercises	50.5 (95.6)	94.0 (119.8)	115.3 (191.7)	113.8 (181.1)	0.0162	0.0173	0.0209

BP = blood pressure, T0 = one week before program, T1 = one week after program, T2 = followed up to 6 months, T3 = followed up to 12 months.

## Data Availability

These study data are deidentified participant data. The data that support the findings of this study are available beginning 12 months and ending 36 months following the article publication from the corresponding author, W-FM, upon reasonable request at lhdaisy@mail.cmu.edu.tw.

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
