# Peer review of "Effects of the Health-Awareness-Strengthening Lifestyle Program in a Randomized Trial of Young Adults with an At-Risk Mental State"

_ijerph, 2021, doi:10.3390/ijerph18041959_

Round 1

Reviewer 1 Report

First of all, I would like to say that I am very thankful to have the opportunity to read this study. The suggestions given in this document are intended to improve your work. If you do not agree with any of them, please explain them to me, and we will try to reach a consensus. The same feedback document will be given to both editors and authors.

Affiliations:

  • Some of the authors’ affiliations are duplicated.

Methods and results:

  • Participants:
    • Inclusion and exclusion criteria should be clearer defined.
    • Lines 104-107 – should this be included in the introduction section?
  • Procedure:
    • Thank you for the HASL program description. The authors provide information about the first, the second and the last session, what about 3,4 and 5th? It should be an interesting information. The full program is always one-to-one? Is there some more information to provide that should ensure the reproducibility of the study?
    • It would be interesting reporting about the statistical power of the study. Maybe they would find useful the G*Power software:
      • Faul, F., Erdfelder, E., Lang, A.-G., & Buchner, A. (2007). G*Power 3: A flexible statistical power analysis program for the social, behavioral, and biomedical sciences. Behavior Research Methods, 39, 175-191.
      • Faul, F., Erdfelder, E., Buchner, A., & Lang, A.-G. (2009). Statistical power analyses using G*Power 3.1: Tests for correlation and regression analyses. Behavior Research Methods, 41, 1149-1160.
      • https://www.psychologie.hhu.de/arbeitsgruppen/allgemeine-psychologie-und-arbeitspsychologie/gpower.html
    • In addition, it would be itnereting to provide a statistical measure of the magnitude of effect of the difference between the groups, such as Cohen's d.

Author Response

Response Letter (Manuscript ID: ijerph-1049017)

To The Reviewer:

Thank you very much for taking time from your busy schedule to comment on our manuscript. Our responses to your suggestions and questions are summarized below, with reference to the appropriate pages in the text. Revisions in the text have also been highlighted in RED. We truly appreciate your thoughtful and constructive comments that help make this a better paper. We are very grateful to you for all the comments and insightful suggestions that help enhance both the quality and readability of our paper. Thank you.

Comments and Suggestions for Authors

First of all, I would like to say that I am very thankful to have the opportunity to read this study. The suggestions given in this document are intended to improve your work. If you do not agree with any of them, please explain them to me, and we will try to reach a consensus. The same feedback document will be given to both editors and authors.

Affiliations:

1.Some of the authors’ affiliations are duplicated.

Response:

Thank you for pointing this out, we had made the corresponded changes (please see line 9-14).

Methods and results: Participants:

2.Inclusion and exclusion criteria should be clearer defined.

Response:

Thank you for your suggestion, we have made the inclusion and exclusion criteria more clear (please see line 119-130).

3.Lines 104-107 – should this be included in the introduction section?

Response:

Thank you for your insightful suggestion, we have included the original Lines 104-107 into the introduction (please see line 68-71).

4.Procedure:

Thank you for the HASL program description. The authors provide information about the first, the second and the last session, what about 3,4 and 5th? It should be an interesting information. The full program is always one-to-one? Is there some more information to provide that should ensure the reproducibility of the study?

Response:

Thank you for pointing this out, and yes, the full program is always one-to-one. We incorporated exercise, stress management, and interpersonal support together into first session; nutrition alone into second session; and self-actualization and health responsibility into third session. Therefore, a total of three sessions. The detailed procedures of HASL program are shown in Table 1 (please see line 166-167).

5.It would be interesting reporting about the statistical power of the study. Maybe they would find useful the G*Power software:

  • Faul, F., Erdfelder, E., Lang, A.-G., & Buchner, A. (2007). G*Power 3: A flexible statistical power analysis program for the social, behavioral, and biomedical sciences. Behavior Research Methods, 39, 175-191.
  • Faul, F., Erdfelder, E., Buchner, A., & Lang, A.-G. (2009). Statistical power analyses using G*Power 3.1: Tests for correlation and regression analyses. Behavior Research Methods, 41, 1149-1160.
  • https://www.psychologie.hhu.de/arbeitsgruppen/allgemeine-psychologie-und-arbeitspsychologie/gpower.html
  • In addition, it would be itnereting to provide a statistical measure of the magnitude of effect of the difference between the groups, such as Cohen's d.

Response:

     Thank you for bringing our attention to this issue. The statistical power of this study is shown below by G*Power software as recommended:

     The participants were divided into experimental group and control group. The estimated total sample size is 82. Considering 20% of lost to follow-up, the estimated number of 99 participants or 50 in each groups can reach a statistical power of 0.8. We had 46 participants in each groups which was very close to 50. We understand the sample size may not enough, so we added small sample size to our limitation (please see line 342).

Reviewer 2 Report

I would like to congratulate the authors for the interesting study both in its research dimension and its social implications.

-Given the large number of studies linking active lifestyles with mental health, perhaps the authors could consider making these aspects more specific in the introduction.

- In Figure 1, it may be more appropriate to reduce the description to "Research flowchart". I understand that it is not necessary to say "This is a figure of".

- Lines 121-125 state that "The program would only be provided to the control group passively, by request, after the post-test. The experimental group was followed up with at 6 months (Q2) and 12 months (Q3) after the program to assess its prolonged effects". Could it be understood that some control group participants might have followed the HASL program before Q2 or Q3? I suppose it would be a misinterpretation, but if so, you may be introducing bias into the T2 and T3 results of the control group.

- Line 128-130: The HASL program is very well-rounded, and I believe that it emphasizes the personal responsibility dimension of a healthy lifestyle. But given that lifestyles are strongly conditioned by the contexts in which humans interact, perhaps a brief reference could be made to the fact that the program does not contemplate this ecological dimension of all socio-health phenomena, as noted in works such as:

-Bann, D., Johnson, W., Li, L., Kuh, D., & Hardy, R. (2018). Socioeconomic inequalities in childhood and adolescent body-mass index, weight, and height from 1953 to 2015: an analysis of four longitudinal, observational, British birth cohort studies. The Lancet Public Health, 3(4), e194–e203.

-Dorado Martín, J. J., Casado Blanco, M., Peral Pacheco, D., Montes Salas, G., Ravelo Antelo, C., Álvarez Gallego, Y., & García Díaz, G. (2005). Relación de la artrosis con el índice de masa corporal y la clase social. Revista Española de Enfermedades Metabólicas Óseas, 14(3), 41–45.

- Higgs, S., & Thomas, J. (2016). Social influences on eating. Current Opinion in Behavioral Sciences, 9, 1–6

- Kohl, H. W.; Craig,  C.L.; Lambert, E. V. et al. (2012). The pandemic of physical inactivity: global action for public health. Lancet. 380: 294-305

- Lusk, J. L., & Ellison, B. (2013). Who is to blame for the rise in obesity? Appetite, 68, 14–20

- Popkin, B.M. & Hawkes, C. (2016). Sweetening of the global diet, particularly beverages: patterns, trends, and policy responses. Lancet Diabetes Endocrinol. 4: 174-186

-Tanumihardjo, S. A., Anderson, C., Kaufer-Horwitz, M., Bode, L., Emenaker, N. J., Haqq, A. M., … Stadler, D. D. (2007). Poverty, Obesity, and Malnutrition: An International Perspective Recognizing the Paradox. Journal of the American Dietetic Association, 107(11), 1966–1972.

 For example, Popkin & Hawkes(2016) points out that: Changes in diet and physical activity reflect changes in demand that are only partly attributable to individual behaviour. The drivers of demand include forces that shape these behaviours, including intensive advertising of less healthful processed foods and sugary drinks, shifts in complementary foods for infants, increased numbers of food retailers that have increased the availability of processed foods, urban design that restricts active transport, and increased numbers of labour-saving devices in homes and workplaces. Another series of drivers underlie these forces, such as international trade agreements, the power of transnational corporations, subsidies that enable the production of inexpensive foods, and government policies or lack thereof related to food taxes or urban design. Mapping of policy changes that affect food9 or physical activity systems10 against shifts in the country or regional prevalence of obesity, undernutrition, and stunting in association with obesity might help identify the most powerful drivers of these changes and begin to point to solutions.

-It would be interesting to know more specific details about the program. Perhaps through a link or by attaching complementary material with the description of the program.

-I believe that section 3.1. ("Sample") could be included in the methodology section, after section 2.2. ("Participants").

Author Response

Response Letter (Manuscript ID: ijerph-1049017)

To the Reviewer:

Thank you very much for taking time from your busy schedule to comment on our manuscript. Our responses to your suggestions and questions are summarized below, with reference to the appropriate pages in the text. Revisions in the text have also been highlighted in RED. We truly appreciate your thoughtful and constructive comments that help make this a better paper. We are very grateful to you for all the comments and insightful suggestions that help enhance both the quality and readability of our paper. Thank you.

Comments and Suggestions for Authors

I would like to congratulate the authors for the interesting study both in its research dimension and its social implications.

1.Given the large number of studies linking active lifestyles with mental health, perhaps the authors could consider making these aspects more specific in the introduction.

Response:

Thank you for your suggestion, we have added more information in the introduction (please see line 88-94).

  1. In Figure 1, it may be more appropriate to reduce the description to "Research flowchart". I understand that it is not necessary to say "This is a figure of".

Response:

Thank you for pointing this out, we have rephrased it into just "Research flowchart" (please see line 111).

  1. Lines 121-125 state that "The program would only be provided to the control group passively, by request, after the post-test. The experimental group was followed up with at 6 months (Q2) and 12 months (Q3) after the program to assess its prolonged effects". Could it be understood that some control group participants might have followed the HASL program before Q2 or Q3? I suppose it would be a misinterpretation, but if so, you may be introducing bias into the T2 and T3 results of the control group.

Response:

Thank you for bringing up this issue. The follow-ups were only being conducted on the experimental group only. This was because the individuals with ARMS have higher risk of developing first episode of psychosis within one to two years; and 18-36% may progress to full psychotic disorder within three years. It is unethical not to provide the intervention to the control group. However, once the control group received the HASL program passively by request after the post-test; they became no different than the experimental group during follow-ups. Therefore, the follow-ups were only being conducted on the experimental group only. We had put this as one of our limitations (please see line 338-342).

  1. Line 128-130: The HASL program is very well-rounded, and I believe that it emphasizes the personal responsibility dimension of a healthy lifestyle. But given that lifestyles are strongly conditioned by the contexts in which humans interact, perhaps a brief reference could be made to the fact that the program does not contemplate this ecological dimension of all socio-health phenomena, as noted in works such as:

-Bann, D., Johnson, W., Li, L., Kuh, D., & Hardy, R. (2018). Socioeconomic inequalities in childhood and adolescent body-mass index, weight, and height from 1953 to 2015: an analysis of four longitudinal, observational, British birth cohort studies. The Lancet Public Health, 3(4), e194–e203.

-Dorado Martín, J. J., Casado Blanco, M., Peral Pacheco, D., Montes Salas, G., Ravelo Antelo, C., Álvarez Gallego, Y., & García Díaz, G. (2005). Relación de la artrosis con el índice de masa corporal y la clase social. Revista Española de Enfermedades Metabólicas Óseas, 14(3), 41–45.

- Higgs, S., & Thomas, J. (2016). Social influences on eating. Current Opinion in Behavioral Sciences, 9, 1–6

- Kohl, H. W.; Craig,  C.L.; Lambert, E. V. et al. (2012). The pandemic of physical inactivity: global action for public health. Lancet. 380: 294-305

- Lusk, J. L., & Ellison, B. (2013). Who is to blame for the rise in obesity? Appetite, 68, 14–20

- Popkin, B.M. & Hawkes, C. (2016). Sweetening of the global diet, particularly beverages: patterns, trends, and policy responses. Lancet Diabetes Endocrinol. 4: 174-186

-Tanumihardjo, S. A., Anderson, C., Kaufer-Horwitz, M., Bode, L., Emenaker, N. J., Haqq, A. M., … Stadler, D. D. (2007). Poverty, Obesity, and Malnutrition: An International Perspective Recognizing the Paradox. Journal of the American Dietetic Association, 107(11), 1966–1972.

 For example, Popkin & Hawkes (2016) points out that: Changes in diet and physical activity reflect changes in demand that are only partly attributable to individual behaviour. The drivers of demand include forces that shape these behaviours, including intensive advertising of less healthful processed foods and sugary drinks, shifts in complementary foods for infants, increased numbers of food retailers that have increased the availability of processed foods, urban design that restricts active transport, and increased numbers of labour-saving devices in homes and workplaces. Another series of drivers underlie these forces, such as international trade agreements, the power of transnational corporations, subsidies that enable the production of inexpensive foods, and government policies or lack thereof related to food taxes or urban design. Mapping of policy changes that affect food9 or physical activity systems against shifts in the country or regional prevalence of obesity, undernutrition, and stunting in association with obesity might help identify the most powerful drivers of these changes and begin to point to solutions.

Response:

Thank you for your knowledgeable suggestion, we have included some of the references above into our references.

References 39 - Kohl, H. W.; Craig,  C.L.; Lambert, E. V. et al. (2012). The pandemic of physical inactivity: global action for public health. Lancet. 380: 294-305

References 40 - Popkin, B.M. & Hawkes, C. (2016). Sweetening of the global diet, particularly beverages: patterns, trends, and policy responses. Lancet Diabetes Endocrinol. 4: 174-186

5.It would be interesting to know more specific details about the program. Perhaps through a link or by attaching complementary material with the description of the program.

Response:

Thank you for pointing this out, the detailed procedures of HASL program are shown in Table 1 (please see line 166-167).

6.I believe that section 3.1. ("Sample") could be included in the methodology section, after section 2.2. ("Participants").

Response:

Thank you for your insightful suggestion, we have included the original section 3.1. ("Sample") in the methodology section after section 2.2. ("Participants") (please see line 132-139).

Reviewer 3 Report

Effects of the health-awareness-strengthening lifestyle program in a randomized trial of young adults with an at-risk mental state

Thank you for asking me to review this manuscript. This randomized trial examined the effects of the health-awareness-strengthening lifestyle (HASL) program among 92 young adults with at-risk mental state in Taiwan. As a health promotion intervention, it might be less stigmatized and beneficial to improve health awareness and help-seeking behaviors. Such an effort could be valuable for future implication, and this manuscript is well-organized. The following issues need to be considered:

In abstract,

  1. About Methods, please provide the description of the control group and the dosage of the HASL program, such as: one-on-one, three 60-90-minute times on the basis of every six weeks.
  2. About Results, were the following effects measured after completing the HASL program for those in experimental group, rather than just participating the program?

“Compared to the control group, those in experimental group showed significantly improvements 41 on anxiety level, health promotion lifestyles, and quality of life one week after participating the program.”

In Introduction,

  1. To strengthen the evidence or knowledge related to the intervention mechanism or theoretical-basis for health promotion in ASRM would be expected to support the HASL program.

From Line 76-94, in general, studies had supported that the effects of health promotion interventions in different population, including healthy people and patients with chronic illness and psychiatric disorders. However, the reader may wonder what seems specific to ARMS different from the other groups, and how it would be? For example, physical health promotion strategies, strengthening one’s health-awareness, knowledge relevant to mental health symptoms and treatment. Have these specific elements tested and targeted for ASRM? What age of the subjects have been targeted in these health promotion interventions for the ASRM?  

  1. Please reconsider the word “desperately” or the sentence of the writing style Line 66-67, avoiding to arbitrary overemphasize the professional assistance since the self-initiated health promotion lifestyle strategies have been addressed on Line 81-82. In particular, both references [8, 9] cited were from the same author group “Carney et al”.

“Thus, individuals with ARMS are the population desperately needing professional helps both physically and psychologically [8, 9]”.

In Methods,

  1. Provide the justification for the ages of the purposive sample ranged from 20-35 years or regarding it as the limitation. Are they too old to receive the early interventions, such as HASL program, compared to the other evidence for ASRM early-intervention studies?
  2. More information is expected to describe the control group, random assignment, and controlling confounding factors between two groups.

--Are both groups received Treatment as Usual? Why only experimental group receive the follow-ups?

--Add information about how to do the random assignment.

--In Line 122, “The program would only be provided to the control group passively, by request, after the post-test.” It might be ethical consideration. But how to control the confounding factors between two groups, for example, they may come from the same clinic or university or class? And how to deal with the John Henry or Hawthorne effects in two groups? Was any blindness included?

  1. Please describe how to deal with the missing data in the longitudinal follow-up in Data analysis.

In Discussions,

  1. Comparing the effects of the HASL program with the other similar studies for ASRM, such as Carney’s group (Reference 8,9) is expected, rather than comparing with those taking anti-psychotics. Implications for developing early interventions in this vulnerable population are suggested, and add suggestions how to translate such empirical findings into clinical practice.
  2. The reasons or difficulties about why the control groups had not been followed up need to be explained in Limitation.

Author Response

Response Letter (Manuscript ID: ijerph-1049017)

To the Reviewers:

Thank you very much for taking time from your busy schedule to comment on our manuscript. Our responses to your suggestions and questions are summarized below, with reference to the appropriate pages in the text. Revisions in the text have also been highlighted in RED. We truly appreciate your thoughtful and constructive comments that help make this a better paper. We are very grateful to you for all the comments and insightful suggestions that help enhance both the quality and readability of our paper. Thank you.

Comments and Suggestions for Authors

Effects of the health-awareness-strengthening lifestyle program in a randomized trial of young adults with an at-risk mental state

 Thank you for asking me to review this manuscript. This randomized trial examined the effects of the health-awareness-strengthening lifestyle (HASL) program among 92 young adults with at-risk mental state in Taiwan. As a health promotion intervention, it might be less stigmatized and beneficial to improve health awareness and help-seeking behaviors. Such an effort could be valuable for future implication, and this manuscript is well-organized. The following issues need to be considered:

In abstract,

  1. About Methods, please provide the description of the control group and the dosage of the HASL program, such as: one-on-one, three 60-90-minute times on the basis of every six weeks.

Response:

Thank you for your insightful suggestion, we have included the description of the control group and the dosage of the HASL program (please see line 39-41).

  1. About Results, were the following effects measured after completing the HASL program for those in experimental group, rather than just participating the program?

“Compared to the control group, those in experimental group showed significantly improvements on anxiety level, health promotion lifestyles, and quality of life one week after participating the program.”

Response:

Thank you for pointing this out, the assessments were done one week after (T1) the program to evaluate its short-term effects.

In Introduction,

  1. To strengthen the evidence or knowledge related to the intervention mechanism or theoretical-basis for health promotion in ASRM would be expected to support the HASL program.

From Line 76-94, in general, studies had supported that the effects of health promotion interventions in different population, including healthy people and patients with chronic illness and psychiatric disorders. However, the reader may wonder what seems specific to ARMS different from the other groups, and how it would be? For example, physical health promotion strategies, strengthening one’s health-awareness, knowledge relevant to mental health symptoms and treatment. Have these specific elements tested and targeted for ASRM? What age of the subjects have been targeted in these health promotion interventions for the ASRM?

Response:

Thank you for pointing this out, there were some researches such as Carney et al. (2016) had introduced a physical health promotion strategy and guideline for the ultra-high risk cohort. They stated that the model provides a systematic approach for developing an intervention and allows theoretically-based BCT to guide behavior change. However, further researches are still needed to put the model into practice. The reason why we conducted this study was because the traditional interventions for ARMS are usually either very costly (psychotherapy) or can produce high social stigma (pharmacotherapy). Health promotion is considered to be a vital strategy for improving health and managing symptoms. It is also proven to be successful in many different populations. Therefore, we wanted to implement a health promotion just for ARMS and accessed its effectiveness.

  1. Please reconsider the word “desperately” or the sentence of the writing style Line 66-67, avoiding to arbitrary overemphasize the professional assistance since the self-initiated health promotion lifestyle strategies have been addressed on Line 81-82. In particular, both references [8, 9] cited were from the same author group “Carney et al”.

“Thus, individuals with ARMS are the population desperately needing professional helps both physically and psychologically [8, 9]”.

Response:

     Thank you for bringing our attention to the writing style, we have rephrased the word “desperately” into “particularly” (please see line 74-76).

In Methods,

  1. Provide the justification for the ages of the purposive sample ranged from 20-35 years or regarding it as the limitation. Are they too old to receive the early interventions, such as HASL program, compared to the other evidence for ASRM early-intervention studies?

Response:

Thank you for pointing this out, according to Nelson et al. (2012), one of the criteria for ultra high risk or ARMS is between the age of 15 and 25. However, due to the legal age in Taiwan is 20 years old and this study is on young adults. Therefore, we chose the ages ranged from 20-35 years to have higher range of recruitment other than 20-25.

Nelson, B., Thompson, A., & Yung, A. R. Basic self-disturbance predicts psychosis onset in the ultra high risk for psychosis "prodromal" population. Schizophrenia bulletin, 38(6), 1277–1287. https://doi.org/10.1093/schbul/sbs007 (2012).

  1. More information is expected to describe the control group, random assignment, and controlling confounding factors between two groups.

--Are both groups received Treatment as Usual? Why only experimental group receive the follow-ups?

Response:

Thank you for bringing up this issue. The program would only be provided to the control group passively by request after the post-test due to ethical reason. This was because the individuals with ARMS have higher risk of developing first episode of psychosis within one to two years; and 18-36% may progress to full psychotic disorder within three years. It is unethical not to provide the intervention to the control group. However, once the control group received the HASL program passively by request after the post-test; they became no different than the experimental group during follow-ups. Therefore, the follow-ups were only being conducted on the experimental group only. We had put this as one of our limitations (please see line 338-342).

  1. Add information about how to do the random assignment.

Response:

Thank you for pointing this out, random assignment was done by computer coding (please see line 141-143).

  1. In Line 122, “The program would only be provided to the control group passively, by request, after the post-test.” It might be ethical consideration. But how to control the confounding factors between two groups, for example, they may come from the same clinic or university or class?

Response:

Thank you for pointing this out, our participants were recruited from different university counseling centers, outpatient psychiatric clinics, and community counseling centers. There was a high percentage that they did not have closed contact with each other. It would be difficult for them to share their treatment or intervention with the people from the same clinic or university or class.

  1. And how to deal with the John Henry or Hawthorne effects in two groups? Was any blindness included?

Response:

Thank you for bringing up the issue of John Henry or Hawthorne effects in two groups. During the recruitment, we let all the participants know that they will received the intervention. However, we did not let them know at which time period. The control group will receive the HASL program only passively after the post-test by request with no follow-ups. The participants will not know which group they are in. Therefore, the participants were blinded.

  1. Please describe how to deal with the missing data in the longitudinal follow-up in Data analysis.

Response:

Thank you for bringing up this issue. We checked participants’ responses very careful every times after they finished the questionnaires. If still having missing data found, the missing data such as lost to follow-up will just be deleted.

In Discussions,

  1. Comparing the effects of the HASL program with the other similar studies for ASRM, such as Carney’s group (Reference 8,9) is expected, rather than comparing with those taking anti-psychotics.

Response:

     Thank you for bringing our attention to the studies by Carney et al., they introduced a physical health promotion strategy including the concepts of the COM-B model and behavior-change wheel. The behavior-change wheel is consisted of three layers. The center layer is known as the COM-B model which consists of motivation, opportunity, and capability. It helps to identify the sources of behavior. The middle layer are the intervention functions include the possible methods to promote behavior change, such as education, persuasion, training, etc. And the outer layer are the policy categories showing how the intervention functions can be applied in the larger scale, for instance, guidelines, regulation, legislation, etc. As for the ultra-high risk cohort, Carney et al. (2016) stated that the model provides a systematic approach for developing an intervention and allows theoretically-based BCT to guide behavior change. Currently, it is still a guideline, further researches are still needed to put the model into practice.

  1. Implications for developing early interventions in this vulnerable population are suggested, and add suggestions how to translate such empirical findings into clinical practice.

Response:

Thank you for pointing this out, we have included our suggestions in our conclusions (please see line 351-357).

  1. The reasons or difficulties about why the control groups had not been followed up need to be explained in Limitation.

Response:

Thank you for your suggestion, we have included the reasons why we did not follow up the control groups in limitation (please see line 338-342). 

Reviewer 4 Report

I agree with the opinion that he health promotion in mental health, especially in relation to early detection and intervention, has become one of a key global health issue. Hypothetically, we can say that according to WHO (2019), health promotion is an activity that encourages people to adopt and maintain healthy lifestyles, and creates supportive living conditions and healthy environments.

This study aimed to explore the short-term and prolonged effects of the health-awareness-strengthening lifestyle (HASL) program, a health promotion lifestyle program that emphasizes self-awareness of mental health risk for the young Taiwanese adults with ARMS. The concept of ARMS emphasized more on negative symptoms and anxiety. This was because negative symptoms, such as schizotypal traits, are statistically predictive factors for the transition from ARMS to mental illness, and high levels of perceived stress and anxiety also occur more frequently in individuals with ARMS than in the general public.

The design of the study and methodology were chosen adequately to address the objectives. A pre-and post-test randomized trial was conducted on young adults with at-risk mental state. The HASL program was provided to the experimental group only. Mental health risk, anxiety level, health promotion lifestyles, quality of life, physiological index, and physical exercises were assessed one week before and after the program and followed up to 6 and 12 months for both groups. The purposive sampling was used to recruit participants between the ages of 20-35 years from university counseling centres, outpatient psychiatric clinics, and community counseling centres.

A strong part of the study is the use of three scales. Three scales were used to identify individuals with higher risk for developing psychosis, including the Chinese Version of the Schizotypal Personality Questionnaire-Brief (CSPQ-B), Chinese Mandarin State and Trait Anxiety Inventory Form Y (CMSTAI-Y), and a demographic inventory to collect information on participants’ family history of mental illness. Individuals who had a confirmed diagnosis of schizophrenia or a mood disorder were excluded.

The results clearly presented: Compared to the control group, those in experimental group showed significantly improvements on anxiety level, health promotion lifestyles, and quality of life one week after participating the program. Furthermore, the experimental group also showed additional long-term positive effect on mental risk, physical exercises, and physical health after the follow-ups.

There is presented a wide discussion. The conclusions supported by the results: The outcomes highlighted the interventions of the HASL program leading to more positive health effects on young adults with at-risk mental state. The implementation of similar clinical service is recommended for young adults with at-risk mental state.

The manuscript is suitable for printing. It is recommended to print without corrections.

Author Response

Response Letter (Manuscript ID: ijerph-1049017)

To the Reviewer:

Thank you very much for taking time from your busy schedule to comment on our manuscript. Thank you.

Comments and Suggestions for Authors

I agree with the opinion that he health promotion in mental health, especially in relation to early detection and intervention, has become one of a key global health issue. Hypothetically, we can say that according to WHO (2019), health promotion is an activity that encourages people to adopt and maintain healthy lifestyles, and creates supportive living conditions and healthy environments.

The design of the study was chosen adequately to address the objectives. A strong part of the study is the use of three scales.

The manuscript is suitable for printing. It is recommended to print without corrections.

Response:

Thank you for your affirmation on our works, it really encourages us to carry on our field of research.
